# Evolution of highly pathogenic H5N1 influenza A virus in the central nervous system of ferrets

Jurre Y. Siegers[1]☯, Lucas Ferreri[2]☯, Dirk Eggink[3], Edwin J. B. Veldhuis Kroeze[1], Aartjan J. W. te Velthuis[4], Marco van de Bildt[1], Lonneke Leijten[1], Peter van Run[1], Dennis de Meulder[1], Theo Bestebroer[1], Mathilde Richard[1], Thijs Kuiken[1], Anice C. Lowen[2], Sander Herfst[1], Debby van Riel [1] *

1 Department of Viroscience, Erasmus MC, Rotterdam, The Netherlands, 2 Department of Microbiology and Immunology, Emory University School of Medicine, Atlanta, Georgia, United States of America, 3 Department of Medical Microbiology, Amsterdam UMC, Amsterdam, The Netherlands, 4 Department of Molecular Biology, Princeton University, Princeton, New Jersey, United States of America

☯ These authors contributed equally to this work.
* d.vanriel@erasmusmc.nl

**Data Availability Statement:** The sequences are available through NCBI's Short Read Archive (https://www.ncbi.nlm.nih.gov/sra/?term= PRJNA911281) BioProject accession number

## Abstract

Central nervous system (CNS) disease is the most common extra-respiratory tract complication of influenza A virus infections in humans. Remarkably, zoonotic highly pathogenic avian influenza (HPAI) H5N1 virus infections are more often associated with CNS disease than infections with seasonal influenza viruses. Evolution of avian influenza viruses has been extensively studied in the context of respiratory infections, but evolutionary processes in CNS infections remain poorly understood. We have previously observed that the ability of HPAI A/Indonesia/5/2005 (H5N1) virus to replicate in and spread throughout the CNS varies widely between individual ferrets. Based on these observations, we sought to understand the impact of entrance into and replication within the CNS on the evolutionary dynamics of virus populations. First, we identified and characterized three substitutions–PB1 E177G and A652T and NP I119M - detected in the CNS of a ferret infected with influenza A/Indonesia/5/ 2005 (H5N1) virus that developed a severe meningo-encephalitis. We found that some of these substitutions, individually or collectively, resulted in increased polymerase activity in vitro. Nevertheless, in vivo, the virus bearing the CNS-associated mutations retained its capacity to infect the CNS but showed reduced dispersion to other anatomical sites. Analyses of viral diversity in the nasal turbinate and olfactory bulb revealed the lack of a genetic bottleneck acting on virus populations accessing the CNS via this route. Furthermore, virus populations bearing the CNS-associated mutations showed signs of positive selection in the brainstem. These features of dispersion to the CNS are consistent with the action of selective processes, underlining the potential for H5N1 viruses to adapt to the CNS.

PRJNA911281. All custom computer code necessary to reproduce the results presented in the manuscript are available on GitHub (https://github.com/genferreri/Evolution-of-highly-pathogenic-H5N1-influenza-A-virus-in-the-central-nervous-system-of-ferrets].

**Funding:** DvR is supported by the Netherlands Organization for Scientific Research (VIDI 91718308) and an EUR fellowship. This work was funded in part by R01 AI154894 to ACL and NIH/NIAID contract HHSN272201400008C to MR and SH. AtV is supported by NIH/NIAID (grants DP2 AI175474-01 and R01 AI170520), and Wellcome Trust (grant 206579/Z/17/Z). The funders had no role in study design, data collection and analysis, decision to publish, or preparation of the manuscript.

**Competing interests:** The authors have declared that no competing interests exist.

## Author summary

The central nervous system (CNS) is one of the most common extra-respiratory tract sites of infection for influenza A viruses. In ferrets—an animal model used to study the pathogenesis of influenza—highly pathogenic avian influenza H5N1 virus can enter the CNS via the olfactory nerve, resulting in the development of a severe meningo-encephalitis. In the present work, we evaluated the evolutionary dynamics of the virus populations entering and spreading throughout the CNS. We show that once inside the CNS, H5N1 viruses can acquire mutations that increase the polymerase activity in vitro. In vivo, the virus bearing these mutations retained its capacity to infect the CNS but showed reduced spread to other anatomical sites. Analysis of virus populations revealed that infection from the nasal turbinate to the olfactory bulb did not present a genetic bottleneck, suggesting a diffusive passage of viruses from the nasal cavity to the CNS. Inside the CNS, specifically in the brainstem, we found signs of positive selection. These findings support the idea that H5N1 viruses can invade the CNS efficiently via the olfactory nerve, and have the potential to adapt to the CNS.

## Introduction

Central nervous system (CNS) disease is the most common and potentially fatal extra-respiratory tract complication of influenza A virus infection [1,2]. The ability of influenza viruses to invade and cause disease in the CNS of humans and other mammals varies widely between virus strains [1,2]. However, infections with highly pathogenic avian influenza (HPAI) H5N1 viruses are often associated with spread to extra-respiratory tissues, including the CNS [3]. Intranasal inoculation of ferrets with HPAI H5N1 viruses often results in infection of the olfactory mucosa and subsequent invasion into the CNS, primarily via the olfactory nerve [2, 4–6] but also via the trigeminal nerve [7]. However, H5N1 virus strains show wide variation in their potential to spread to the CNS [8]. Among individual ferrets infected with the same strain there are also differences in the extent of virus replication and spread throughout the CNS [4,6,9]. The reasons behind this heterogeneity, at both the virus and host levels, are poorly understood.

Once inside the CNS, virus populations encounter cells of neuronal and glial origin that differ phenotypically from cells of the respiratory tract. Viral adaptation to these cells could occur if conditions are conducive to positive selection. A major prerequisite of positive selection is genetic variation. It is therefore relevant to evaluate whether a genetic bottleneck constricts the diversity of viral populations accessing the CNS, which would reduce the potential for adaptation. Whether or not it is subject to a bottleneck, the diversity of the seed population can however be augmented through de novo mutation, resulting from the error-prone replication of the viral genome by the influenza virus RNA-dependent RNA polymerase. The potential for positive selection to lead to adaptation is then further reliant on the extent to which selection and drift are active in the virus population [10]. Selection is a deterministic process in which the relative fitness of a variant leads to changes in its frequency: during positive selection, variants increase in frequency owing to a fitness advantage; during purifying selection, variants decrease in frequency and are purged from the population owing to a fitness defect [11]. Beside selection, genetic drift occurs, which is a stochastic process by chance, independent of the fitness [11]. Whether evolution of influenza A viruses in the CNS occurs is currently unknown. Similarly, whether any evolution is driven by positive selection, leading to adaptation to the new environment has not been explored.

In the present study, we evaluated an H5N1 HPAI virus that acquired three substitutions–PB1 E177G and A652T and NP I119M - in the CNS of an experimentally inoculated ferret in order to evaluate the adaptive potential of H5N1 virus to the CNS. This ferret developed a severe meningo-encephalitis associated with virus spread throughout the CNS, including the olfactory bulb, cerebrum, cerebellum, and spinal cord [6]. We performed a phenotypic characterization, *in vitro* and *in vivo*, of the virus with the CNS-associated mutations (H5N1-CNS virus) and compared it with the wild type virus (H5N1-WT). We found that the H5N1-CNS virus demonstrated higher polymerase activity than the wild type virus *in vitro* but did not show increased replication in the CNS *in vivo*. Analysis of viral variant dynamics within the infected ferrets revealed that viral dissemination into the CNS via the olfactory nerve was not subject to stochastic loss of diversity through genetic bottlenecks. Importantly, the H5N1-CNS virus also showed signs of positive selection in the brainstem, suggesting that H5 HPAI viruses are able to evolve under positive selection within the CNS.

## Results

### H5N1 virus acquired substitutions PB1-177G, PB1-652T and NP-119M within the ferret CNS

In a prior study, an HPAI H5N1 virus inoculated ferret developed a severe meningoencephalitis with viral replication throughout the CNS. To investigate whether the HPAI H5N1 virus acquired mutations in the CNS, viruses were sequenced from the nasal turbinates, cerebellum and cerebrospinal fluid (CSF) at 7 days post infection (dpi) and the inoculum [6]. Sequence analyses identified three amino acid substitutions above a frequency of 1% in the viruses from the cerebellum and CSF samples, PB1 E177G and A652T and NP I119M, which were not present in viruses in the inoculum or nasal turbinates [Table 1]. Additionally, one synonymous mutation was observed in the coding region of HA (nucleotide T66C, coding for G22). The fact that PB1 E177G was fixed and that the frequency of PB1 A652T ranged from 0.53 to 0.66 suggests that occurrence of these variants was not co-dependent. Sequencing of the virus inoculum did not reveal the presence of these variants, suggesting that these mutations occurred de novo within the infected ferret.

To understand the interactions and possible associations between these mutations, we evaluated their position within the ribonucleoprotein complex [Fig 1]. Structural analysis showed that position 177 in PB1 resides at the base of the β-ribbon domain (residues 177–214) [Fig 1A]. The β-ribbon is a hinged, solvent-exposed domain that contains the two nuclear localization signal (NLS) motifs important for RanBP5 binding and nuclear import of the PA-PB1 heterodimer [12,13]. Furthermore, the flexible β-ribbon has been proposed to play a role in viral RNA binding by contacting the 3'-end of the vRNA when bound to the A-side on the exterior of the polymerase complex [13,14].

**Table 1. Mutations detected in the CNS of a ferret infected with virus of the H5N1 subtype.** Three substitutions were found in common between the cerebellum (CL) and cerebrospinal fluid (CSF). REF refers to the nucleotide reference; ALT to the nucleotide alternative variant; REF-AA amino acid reference; ALT-AA to substitution. Sequencing was done using MySeq Illumina platform.

| Sample | Segment | Position | REF | ALT | REF-AA | ALT-AA | Frequency | Coverage |
|--------|---------|----------|-----|-----|--------|--------|-----------|----------|
| CL | PB1 | 554 | A | G | E | G | 0.994 | 5011 |
| CL | PB1 | 1978 | G | A | A | T | 0.662 | 4683 |
| CL | NP | 402 | T | G | I | M | 0.991 | 7980 |
| CSF | PB1 | 554 | A | G | E | G | 0.993 | 7934 |
| CSF | PB1 | 1978 | G | A | A | T | 0.534 | 7930 |
| CSF | NP | 402 | T | G | I | M | 0.992 | 7986 |

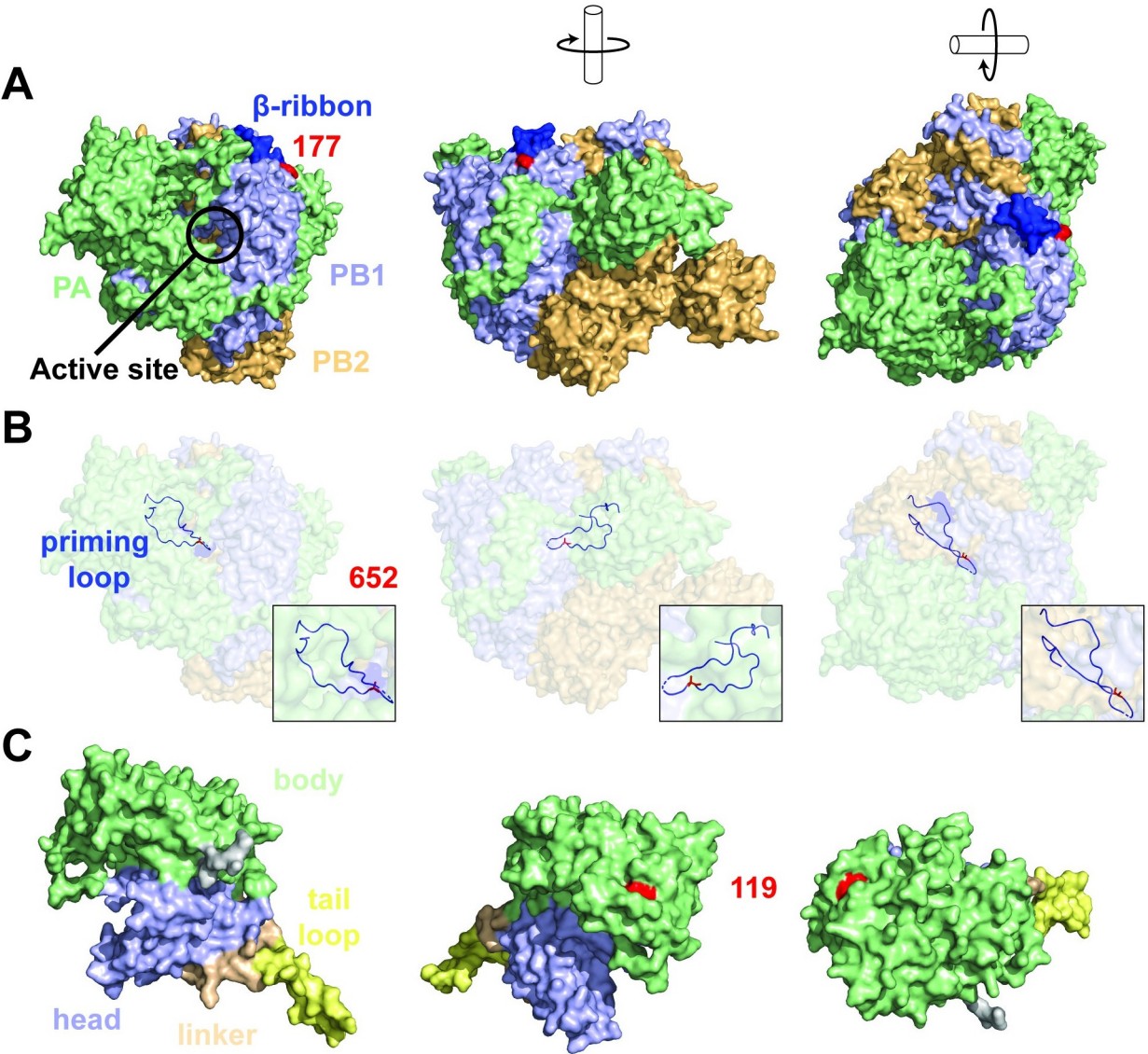

**Fig 1. Location of PB1 E177G and A652T on the polymerase complex and I119M on the nucleoprotein.** A, overview of the polymerase subdomains with PB1 (light-blue), PB2 (paleyellow), and PA (pale-green) in different orientations and the location of PB1 residue 177 (red). B, location of PB1 residue 652 (red) showing individual atoms within the priming loop (light-blue) on a transparent polymerase background. C, overview of the nucleoprotein subdomains with head (light-blue), tail loop (yellow), linker (pale-orange) and body (pale-green) in different orientations and the location of NP residue 119 (red).

The surface electrostatic potential is important for the affinity and specificity of macromolecular interactions, protein folding, and chemical reactivity. [15] For example, the adaptive E627K mutation in the PB2 subunit of the RNA polymerase changes the surface potential and influences the interaction with host factor ANP32A or importins-α isoforms. [16–20]. To investigate whether PB1-E177G could affect a putative interaction with a host or viral binding partner, we evaluated the electrostatic surface potential of the wild-type and CNS mutant PB1 subunit in the unbound (apo) form (S1 Fig) and promotor-bound form of the polymerase [S2 Fig]. In the apo form [S1A and S1B Fig], we observed a change in the surface potential from negative to neutral [S1C and S1D Fig]. In the promotor-bound form [S2A Fig], we observed a

shift from moderately negative to moderately positive charge [S2C and S2D Fig]. These results suggest that the PB1-E177G substitution could influence interactions with potential (host) binding factors in both forms of the polymerase.

Residue PB1-652 resides in the priming loop domain, a β-hairpin structure located near the active site of the RNA polymerase [Fig 1B, S3 Fig] [21,22], which has several key functions in influenza virus RNA synthesis [22]. No specific function for residue 652 has been described yet, but analysis of the RNA polymerase structure suggests that substitutions at position PB1-A652T may affect efficient stabilization of the 3' ssRNA positioning, mediated by residue PB1-651 [22,23] [S3C Fig].

Next, we mapped the location of residue NP-119 using the H5N1 NP X-ray structure [Fig 1C]. Residue NP-119 is located in the body domain and likely not responsible for homodimer formation. Furthermore, superposing the H5N1 NP structure onto the vRNP of A/WSN/33 revealed that residue 119 is likely not involved in any RNA interactions [S4A–S4D Fig]. The relatively large distance of residue 119 with an opposing NP molecule suggests that this residue plays no direct role in NP-NP interactions [S4D Fig]. This is further supported by the electrostatic surface maps showing the large positively charged RNA-biding grove between the head and body domain [S5A–S5D Fig].

## CNS-associated mutations enhance polymerase activity in different human cell types

As substitutions were found in the polymerase complex and NP, mediating transcription and replication of the viral genome [24], we investigated the effect of these substitutions on polymerase activity using a minigenome assay in human kidney cells (HEK293T), human lung cells (A549) and human neuronal cells (SH-SY5Y). The mammalian adaptive substitution PB2-E627K, which increases viral polymerase activity, was included as positive control [25,26]. The single substitution PB1-652T resulted in a significant increase of polymerase activity in A549 and SH-SY5Y cells [Fig 2A, S1 Data]. PB1-177G or NP-119M or a combination of NP-119M with PB1-177G or PB1-652T maintained wild-type levels of polymerase activity. The combined substitutions PB1-177G/652T with or without NP-119M significantly improved transcriptional activity in all cell lines, but most prominently in A549 cells [Fig 2A]. Whereas the combination of the three substitutions enhanced polymerase activity in all cell lines, replication of the virus carrying all three substitutions was significantly increased relative to wild type virus only in MDCK cells [Fig 2B, S2 Data].

## No increased replication of H5N1-CNS virus compared to H5N1-WT in vivo

To evaluate the phenotypes associated with the CNS-associated mutations in vivo, we compared the pathogenesis of H5N1-CNS and H5N1-WT virus in ferrets. Six ferrets were intranasally inoculated with either H5N1-CNS or H5N1-WT virus, followed by euthanasia and tissue sample collection at 3 and 6 dpi. Sneezing was apparent in both groups but no other respiratory signs were observed [Table 2]. Reduced physical activity was observed in both groups and neurological signs were observed in all ferrets except in one ferret from the H5N1-CNS group (Ferret # F11, see Table 3 for ferret numbers).

Weight loss was observed in all ferrets up to 6 dpi, with the exception of ferret F11 from the H5N1-CNS group, which started to gain weight again after 4 dpi [Fig 3A, S3 Data]. At 1 dpi, weight loss of ferrets infected with H5N1-CNS virus (mean [SD], 5.1% [1.9]) was significantly higher compared to ferrets infected with H5N1-WT virus (3% [0.8]).

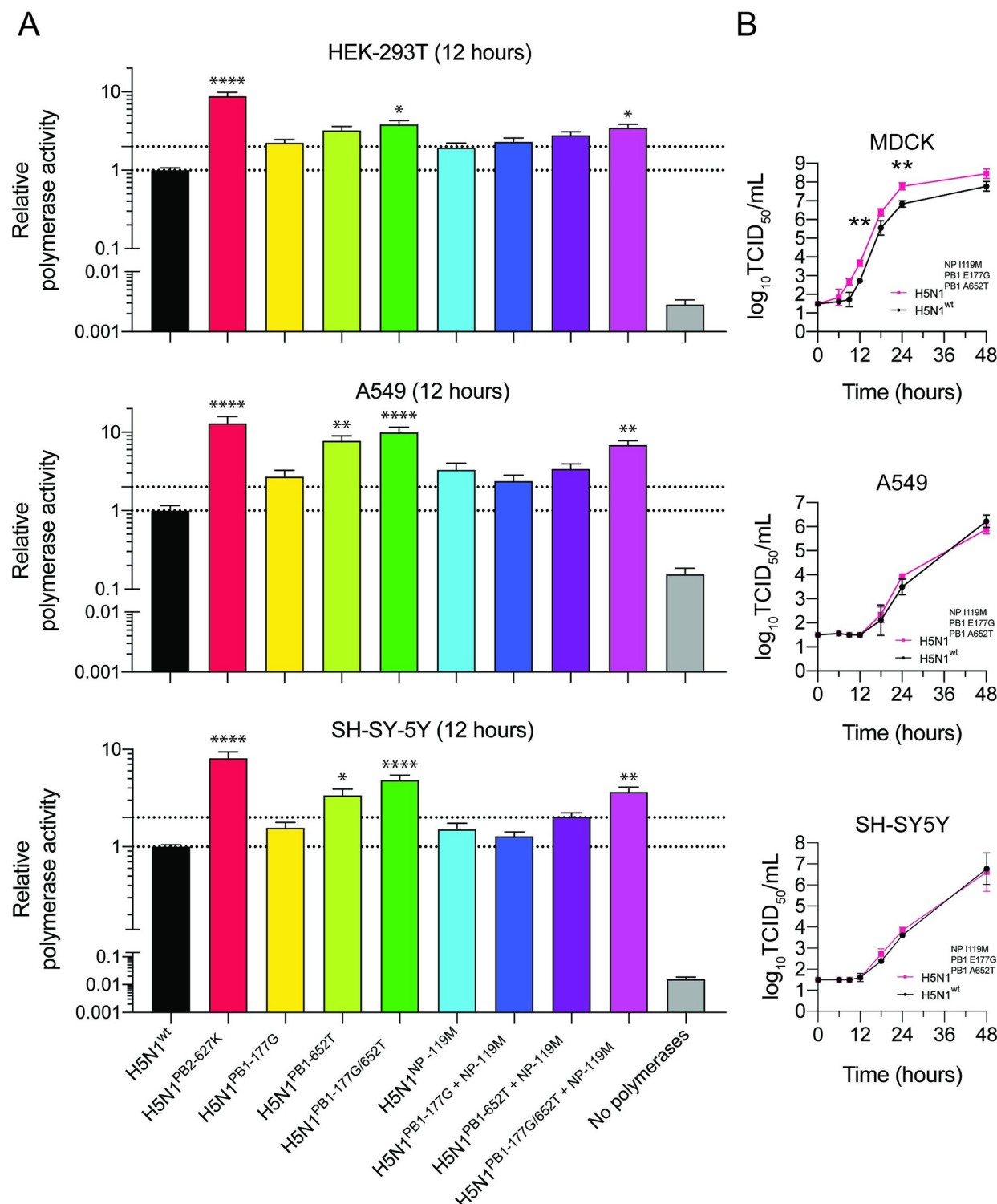

**Fig 2. Polymerase activity of polymerase gene segments carrying PB1 E177G, A652T and NP I119M and in vitro replication kinetics of viruses with these PB1 and NP substitutions.** Activity of polymerase complexes carrying PB1 and NP substitutions in HEK-293T, A549, and SH-SY5Y cells (A). Polymerase activity is presented relative to that of H5N1-WT. Dotted lines indicate polymerase activity at H5N1-WT level and 2-fold increased activity. Mutant PB1 or NP genes that have significantly different polymerase or replicative (B) activity from that of H5N1-WT. Signifancies are indicated by asterisks, identified using a one-way (A) analysis of variance (ANOVA) with Bonferroni's multiple-comparison tests or a two-way (B) analysis of variance (ANOVA) with Dunnett's multiple-comparison tests. *, $P \le 0.05$; **, $P \le 0.01$; ***, $P \le 0.001$; ****, $P \le 0.0001$. Data are presented as means ±SEMs for panels A and ±SDs for panels B from at least three independent experiments.

**Table 2. Clinical signs in H5N1-WT and H5N1-CNS virus inoculated ferrets.**

| Days post inoculation | H5N1-WT (No./Total) | | | | | | | H5N1-CNS (No./Total) | | | | | | |
|---|---|---|---|---|---|---|---|---|---|---|---|---|---|---|
| | 0 | 1 | 2 | 3 | 4 | 5 | 6 | 0 | 1 | 2 | 3 | 4 | 5 | 6 |
| Respiratory signs | | | | | | | | | | | | | | |
| Sneezing | - | - | 6/6 | 6/6 | 3/3 | - | - | - | - | 6/6 | 6/6 | 1/3 | - | - |
| Nasal discharge | - | - | - | - | - | - | - | - | - | - | - | - | - | - |
| Dyspnea | - | - | - | - | - | - | - | - | - | - | - | - | - | - |
| Hunched posture | - | - | - | - | - | - | - | - | 1/3 | - | - | - | - | - |
| Systemic signs | | | | | | | | | | | | | | |
| Physical condition | N/A | N/A | N/A | Good | N/A | N/A | Moderate 1/3 Poor 1/3 Good 1/3 | N/A | N/A | N/A | Good | N/A | N/A | Moderate 1/3 Good 2/3 |
| Activity score* | 0 (6/6) | 0 (6/6) | 0 (6/6) | 0 (6/6) | 0 (3/3) | 1 (3/3) | 2 (2/3) 3 (1/3) | 0 (6/6) | 0 (6/6) | 0 (6/6) | 0 (6/6) | 0 (6/6) | 1 (2/3) 0 (1/3) | 0 (1/3) 1 (1/3) 2 (1/3) |
| Ruffled fur | - | - | - | - | - | - | 3/3 | - | 3/3 | - | - | - | - | 3/3 |
| Neurological signs | | | | | | | | | | | | | | |
| Ataxia | - | - | - | - | - | - | 3/3 | - | - | - | - | 1/3 | - | 2/3 |
| Parases | - | - | - | - | - | - | 3/3 | - | - | - | - | 1/3 | - | 2/3 |
| Tremors | - | - | - | - | - | - | 1/3 | - | - | - | - | - | - | - |

*0: Alert and playful

*1: Alert and playful only when stimulated

*2: Alert but not playful when stimulated

*3: Neither alert nor playful when stimulated

N/A: not applicable

We assessed virus shedding from the respiratory tract by measuring viral titers in throat and nasal swabs. In the throat, virus was detected up to day 6, peaking at 2 dpi (mean [SD] of $10^{6.0 [0.3]}$ TCID$_{50}$/mL for H5N1-WT group and $10^{5.7 [0.8]}$ TCID$_{50}$/mL for H5N1-CNS group) [Fig 3B, S4 Data]. In the nose, virus was detected from both groups of ferrets up to 6 dpi with peak shedding for H5N1-WT at 2 dpi ($10^{2.8 [0.9]}$ TCID$_{50}$/mL) and for H5N1-CNS at 5 dpi ($10^{3.1 [0.8]}$ TCID$_{50}$/mL). At 1 dpi, viral titers in nose swabs from ferrets inoculated with H5N1-WT virus ($10^{3.8 [0.5]}$ TCID$_{50}$/mL) were significantly higher compared to ferrets inoculated with H5N1-CNS virus ($10^{2.1 [0.5]}$ TCID$_{50}$/mL) [Fig 3C, S4 Data].

Macroscopic examination revealed a moderate-to-poor physical condition in two and one ferret from the H5N1-WT and H5N1-CNS groups, respectively at 6 dpi and good physical condition of all other ferrets at 3 and 6 dpi [Table 2]. In tissues from the respiratory tract, the highest viral titers were detected in the nasal turbinates, with no difference between ferrets inoculated with H5N1-WT or H5N1-CNS virus [Fig 3D, S5 Data]. Viral titers were also detected in the soft palate and tonsil of both groups. Viruses were occasionally detected in the trachea, bronchus, lung and tracheobronchial lymph node of ferrets inoculated with H5N1-WT virus, but not in CNS-virus inoculated ferrets.

In tissues of the CNS, virus was detected in the olfactory bulb from both H5N1-WT and H5N1-CNS virus inoculated ferrets at 3 and 6 dpi with significantly higher titers in the olfactory bulb of H5N1-WT virus inoculated ferrets compared to ferrets inoculated with H5N1-CNS virus [Fig 3E, S5 Data]. No significant differences were observed in the viral titers in the cerebrum, cerebellum, and brainstem between groups. For ferret

**Table 3. Virus Antigen Detection in the respiratory tract, central nervous system, and other organs.**

| | 3 dpi | | | | | | 6 dpi | | | | | |
|---|---|---|---|---|---|---|---|---|---|---|---|---|
| | H5N1-WT | | | H5N1-CNS | | | H5N1-WT | | | H5N1-CNS | | |
| Ferret # | F1 | F2 | F3 | F4 | F5 | F6 | F7 | F8 | F9 | F10 | F11 | F12 |
| **Respiratory system** | | | | | | | | | | | | |
| Nasal Turbinates | ++ | + | ++ | ++ | + | + | + | ++ | ++ | + | + | + |
| Soft Palate | - | - | - | - | - | - | - | - | - | - | - | - |
| Trachea | - | - | - | - | - | - | - | - | - | - | - | - |
| Bronchus | - | - | - | - | - | - | - | - | - | - | - | - |
| Lung | - | - | - | - | - | - | - | - | - | - | - | - |
| Tonsil | + | n/a | + | + | n/a | - | - | - | - | - | - | - |
| Tip of nose | +* | - | - | - | - | - | - | - | - | - | - | - |
| **Central nervous system** | | | | | | | | | | | | |
| Olfactory bulb | - | - | - | + | - | - | ++ | + | + | + | + | + |
| Cerebrum | - | - | - | - | - | - | + | + | + | + | - | + |
| Cerebellum | - | - | - | - | - | - | ++ | + | + | + | - | + |
| Brainstem | - | - | - | - | - | - | + | + | - | + | - | + |
| spinal cord (cervical) | n/a | n/a | n/a | n/a | n/a | n/a | + | + | + | + | - | + |
| spinal cord (lumbar) | n/a | n/a | n/a | n/a | n/a | n/a | - | +* | - | - | - | - |
| Trigeminal ganglion | - | - | - | - | - | - | + | + | + | + | - | - |
| **Other** | | | | | | | | | | | | |
| Tongue | - | + | - | - | - | - | - | + | - | + | - | - |
| Heart | - | - | - | - | - | - | - | - | - | - | - | - |
| Liver | - | - | - | - | - | - | - | - | - | - | - | - |
| Spleen | - | - | - | - | - | - | - | - | - | - | - | - |
| Kidney | - | - | - | - | - | - | - | - | - | - | - | - |
| Adrenal | - | - | - | - | - | - | - | - | - | - | - | - |
| Pancreas | - | - | - | - | - | - | + | - | - | - | - | - |
| Jejunum | - | - | - | - | - | - | - | - | - | - | - | - |
| GALT | - | - | + | +* | +* | - | - | - | - | - | - | - |
| Mesenteric LN | n/a | + | + | - | - | - | - | n/a | - | n/a | - | n/a |

- negative

+ postive cells

++ abundant positive cells

* sporadic positive cell

n/a not available

H5N1-CNS F11, virus could only be detected in the olfactory bulb but not in other parts of the CNS.

We also examined the presence of virus in the heart, liver, spleen, adrenal gland, kidney, pancreas, jejunum, blood, serum and plasma. Virus was sporadically detected in the liver, spleen, adrenal, kidney, pancreas, and jejunum of ferrets inoculated with H5N1-WT, but not in ferrets inoculated with H5N1-CNS [S6 Fig].

Together, these data suggest that H5N1-CNS virus retained the capacity to invade and spread throughout the CNS, although virus titers were lower in the olfactory bulb, lower respiratory tract and extra-respiratory organs compare to ferrets inoculated with H5N1-WT.

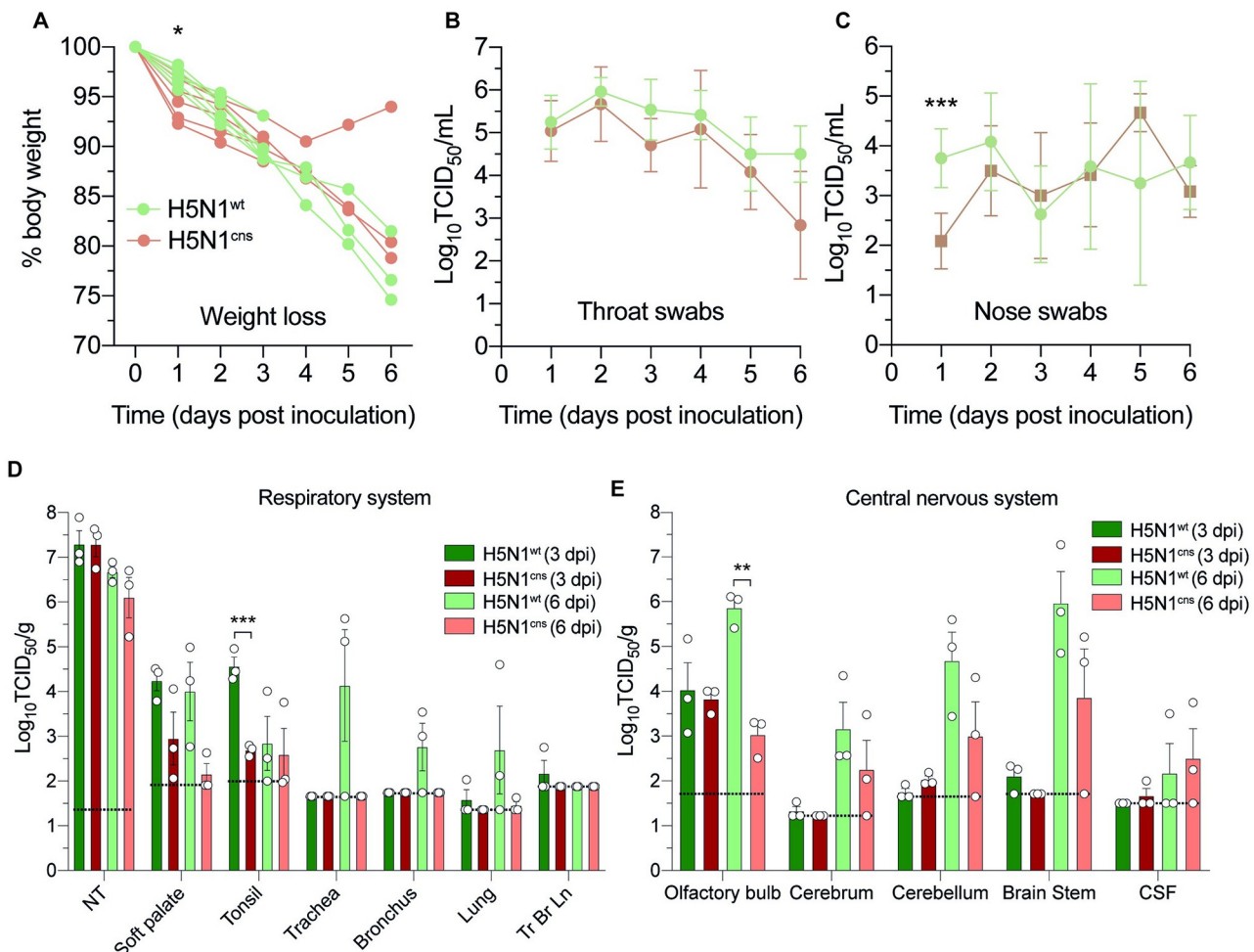

**Fig 3. In vivo experimental infection of ferrets with of H5N1-CNS or WT-virus.** A, weight loss (% of starting body weight) of ferrets inoculated with H5N1-WT or H5N1-CNS virus. Virus titers in throat (B) and nose (C) swabs. D, virus detection in respiratory tract tissues. NT; nasal turbinates, Tr Br Ln; tracheobronchial lymph node. E, virus detection in different regions of the central nervous system. CSF; cerebrospinal fluid. Data are presented as means ±SDs. Statistical analysis was performed using multiple independent unpaired t-tests. TCID; tissue culture infectious dose, *,p ≤ 0.05; **, P ≤ 0.01; ***, p ≤ 0.001. Dotted lines represent the limit of detection.

## Histopathology and distribution of virus antigen in H5N1-CNS and H5N1-WT inoculated ferrets

In the respiratory tract, microscopic examination revealed mild-to-moderate rhinitis and necrosis in the epithelium of the nasal mucosa in all ferrets [Table 3, Fig 4]. In the nasal turbinates, influenza virus antigen was associated with histological lesions and predominantly present in cells of the olfactory mucosa [Fig 4]. In the CNS, virus antigen was detected in the olfactory bulb of one ferret from the H5N1-CNS 3 dpi group [Table 3] without histological lesions. In all ferrets at 6 dpi (except ferret F11), influenza virus antigen was detected in meningeal cells, ependymal cells of ventricles and spinal canal, and neurons in the olfactory bulb, cerebrum, cerebellum and brain stem [Table 3; Fig 4]. Foci of necrosis and inflammation co-localized with virus antigen consistent with an acute mild-to-moderate meningo-encephalitis. Virus antigen was observed in few neuronal cells of the trigeminal ganglia in all ferrets from the H5N1-WT and in one ferret from the H5N1-CNS at 6 dpi [Table 3, Fig 4]. Virus antigen

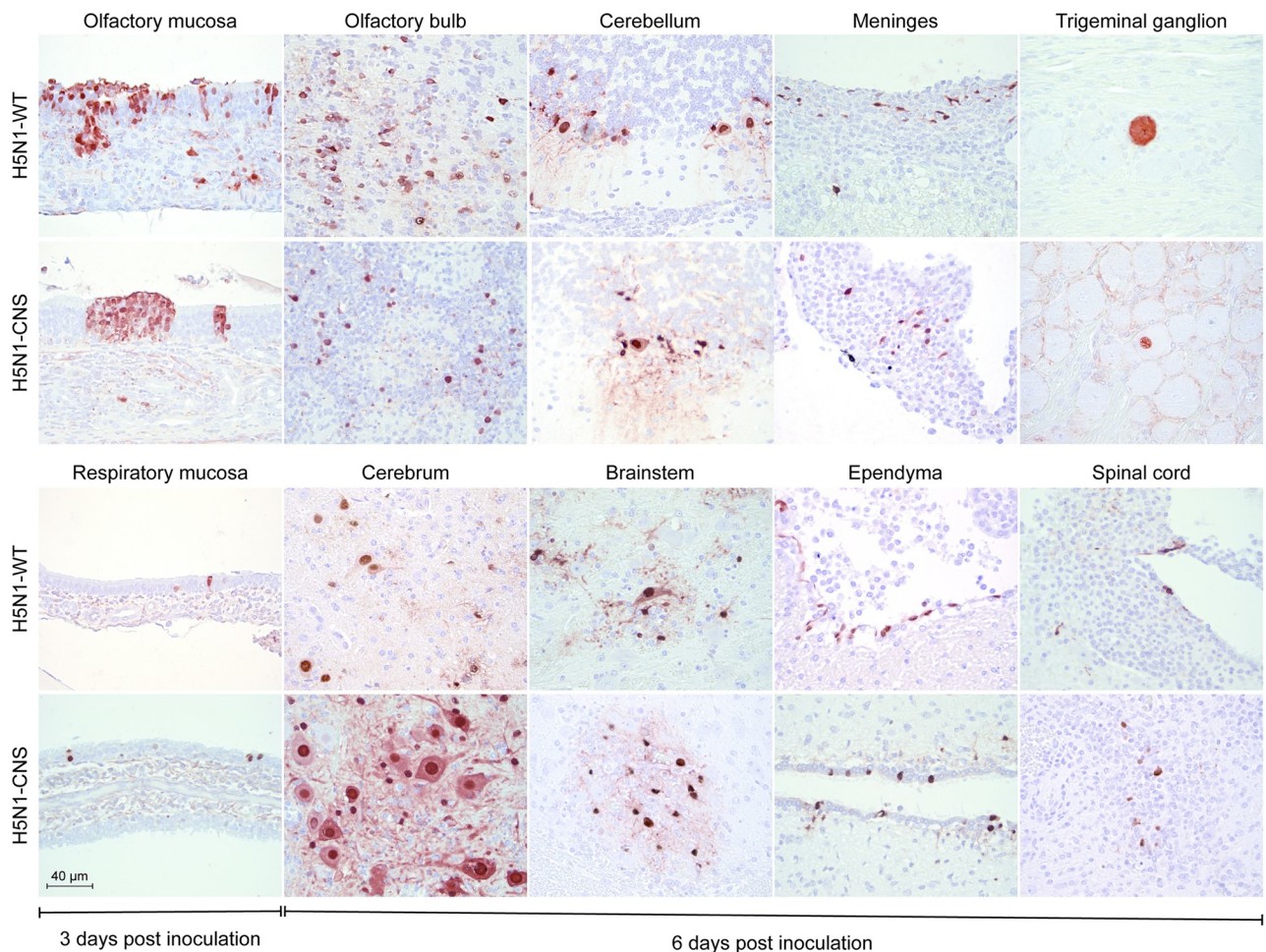

**Fig 4. Virus infected cells detected by immunohistochemistry in the nasal turbintes at 3 days post inoculation and in the CNS at 6 days post inoculation in H5N1-WT or H5N1-CNS virus inoculated ferrets.** Staining for influenza A virus nucleoprotein is seen as red a red precipitate. Representative bright-field images are shown.

was occasionally detected in lymphoid tissues and/or pancreatic epithelium [Table 3]. A detailed pathological description is available in S1 Data.

## Sequence analysis reveals stability of CNS-associated mutations in vivo

To assess the stability of the CNS-mutations in vivo, and to ensure that the H5N1-WT virus did not acquire mutations at these sites during the experiment, viruses from the respiratory tract and CNS were sequenced using the Illumina platform. The original sequences at the sites of the three CNS-associated mutations were maintained in all ferrets (Tables 4 and 5, S1 and S2 Tables), indicating that the variants examined were stable *in vivo*.

## Viral dissemination from the nasal cavity to the CNS along the olfactory nerve is associated with a loose bottleneck

Entering a new environment can impose genetic bottlenecks on virus populations, reducing genetic diversity. If such contractions are considerable, then stochastic founder effects can occur. To assess if virus transmission to the CNS via the olfactory nerve results in a genetic

**Table 4. In vivo stability of molecular signatures for CNS-mutations' sites at 3 dpi.**

| Gene segment | PB1 | | NP |
|---|---|---|---|
| Amino acid | **177** | **652** | **119** |
| Reference | E | A | I |
| Nasal turbinates | | | |
| F1, F2, F3 | E | A | I |
| F4, F5, F6 | **G** | **T** | **M** |
| Olfactory bulb | | | |
| F1, F2, F3 | E | A | I |
| F4, F5, F6 | **G** | **T** | **M** |

bottleneck, we compared viral diversity from H5N1-WT-infected ferrets. We performed the analysis on samples that met a cutoff of $1x10^4$ $TCID_{50}$/g, which included one ferret at 3 dpi and three ferrets at 6 dpi. Contrary to expectation for a stringent bottleneck, we found increased diversity in the olfactory bulb compared to the nasal turbinates in all four animals [Fig 5].

To gain insight into the processes governing the transfer of viruses between tissues, we compared the variants detected in the nasal turbinate and the olfactory bulb of H5N1-WT-inoculated ferrets at 6 dpi [Fig 6]. While the majority of variants were unique to each tissue, we found that, in the olfactory bulb, 3 in 16, 1 in 10 and 4 in 12 variants were also found in the nasal turbinate from ferret F7, F8 and F9, respectively. These common variants were furthermore found at comparable frequencies in the two tissues. Since it is highly unlikely for

**Table 5. In vivo stability of molecular signatures for CNS-mutations' sites at 6 dpi.**

| Gene segment | PB1 | | NP |
|---|---|---|---|
| Amino acid | **177** | **652** | **119** |
| Reference | E | A | I |
| Nasal turbinates | | | |
| F7, F8, F9 | E | A | I |
| F10, F11, F12 | **G** | **T** | **M** |
| Olfactory bulb | | | |
| F7, F8, F9 | E | A | I |
| F10, F11, F12 | **G** | **T** | **M** |
| Cerebrum | | | |
| F7, F8, F9 | E | A | I |
| F10, F11*, F12 | **G** | **T** | **M** |
| Cerebellum | | | |
| F7, F8, F9 | E | A | I |
| F10, F11*, F12 | **G** | **T** | **M** |
| Brainstem | | | |
| F7, F8, F9 | E | A | I |
| F10, F11, F12 | **G** | **T** | **M** |
| CSF | | | |
| F7, F8*, F9 | E | A | I |
| F10, F11*, F12 | **G** | **T** | **M** |

* sequence coverage in viruses was too low and excluded from the analysis.

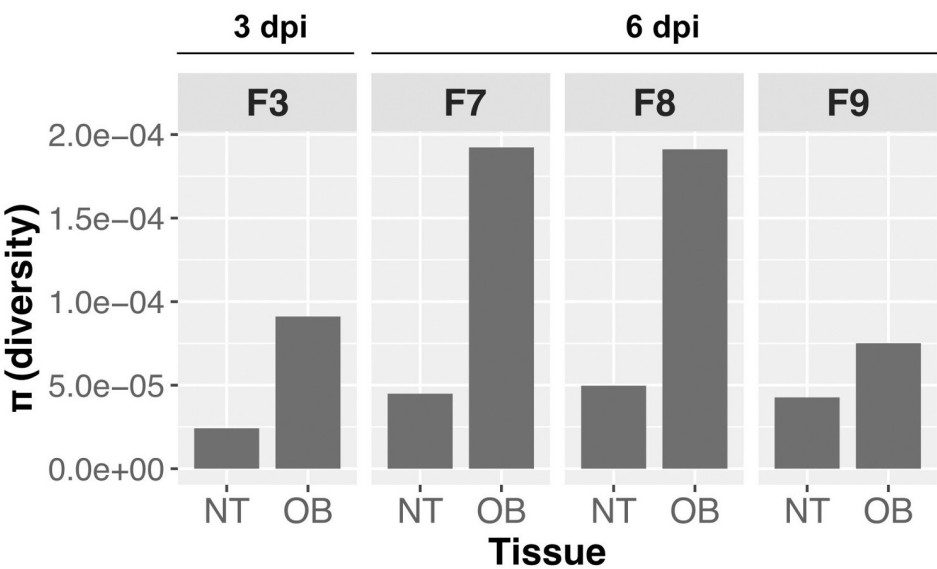

**Fig 5. Absence of bottleneck for H5N1-WT virus entering the CNS via the olfactory nerve.** H5N1-WT-virus infected ferrets show higher overall diversity in the olfactory bulb (OB) than in the nasal turbinate (NT).

common variants at comparable frequencies to arise independently, these data suggest that multiple variants penetrated the CNS from the nasal turbinate. Thus, bottlenecks may not play a role during viral dissemination between these sites. The unique variants identified at each site suggest that, in addition to transfer of multiple variants between the sampled locations, posterior diversification within the olfactory bulb or spread of unsampled populations from the nasal turbinates to this site occurred.

## Dissemination of virus populations through the CNS

To further evaluate population dispersion within the CNS, we evaluated the commonness and uniqueness of variants present in the nasal turbinates, olfactory bulb and brainstem. From the H5N1-WT virus, 2 in 10, 1 in 5 and 2 in 12 variants from the nasal turbinate were found in the brainstem of ferrets F7, F8 and F9, respectively. Variants shared between the olfactory bulb and the brainstem were 4 in 10, 2 in 5 and 1 in 13 for ferrets F7, F8 and F9, respectively [Fig 6]. Analysis of the nasal turbinate and brainstem from ferrets inoculated with the H5N1-CNS virus showed that 2 of 14 were shared for ferret F10, whereas ferret F12 presented only unique variants [S7 Fig]. These results suggest that dissemination of variants through the CNS occurs but that *in situ* differentiation of viral populations is the primary driver of diversification.

## Independent H5N1 virus evolution drives positive selection within the CNS

Virus populations facing new environments or with a fitness defect are subject to fitness increment through positive selection. To evaluate for signs of positive selection at the different anatomical locations, we calculated the ratio πN/πS. Owing to the greater fitness effects of nonsynonymous changes compared to synonymous ones, populations under positive selection typically show ratios above 1.0.

In nasal turbinates collected at 3 dpi, we did not detect signs of positive selection [S8 Fig] for H5N1-WT or H5N1-CNS viruses. However, at 6 dpi, evidence of positive selection acting

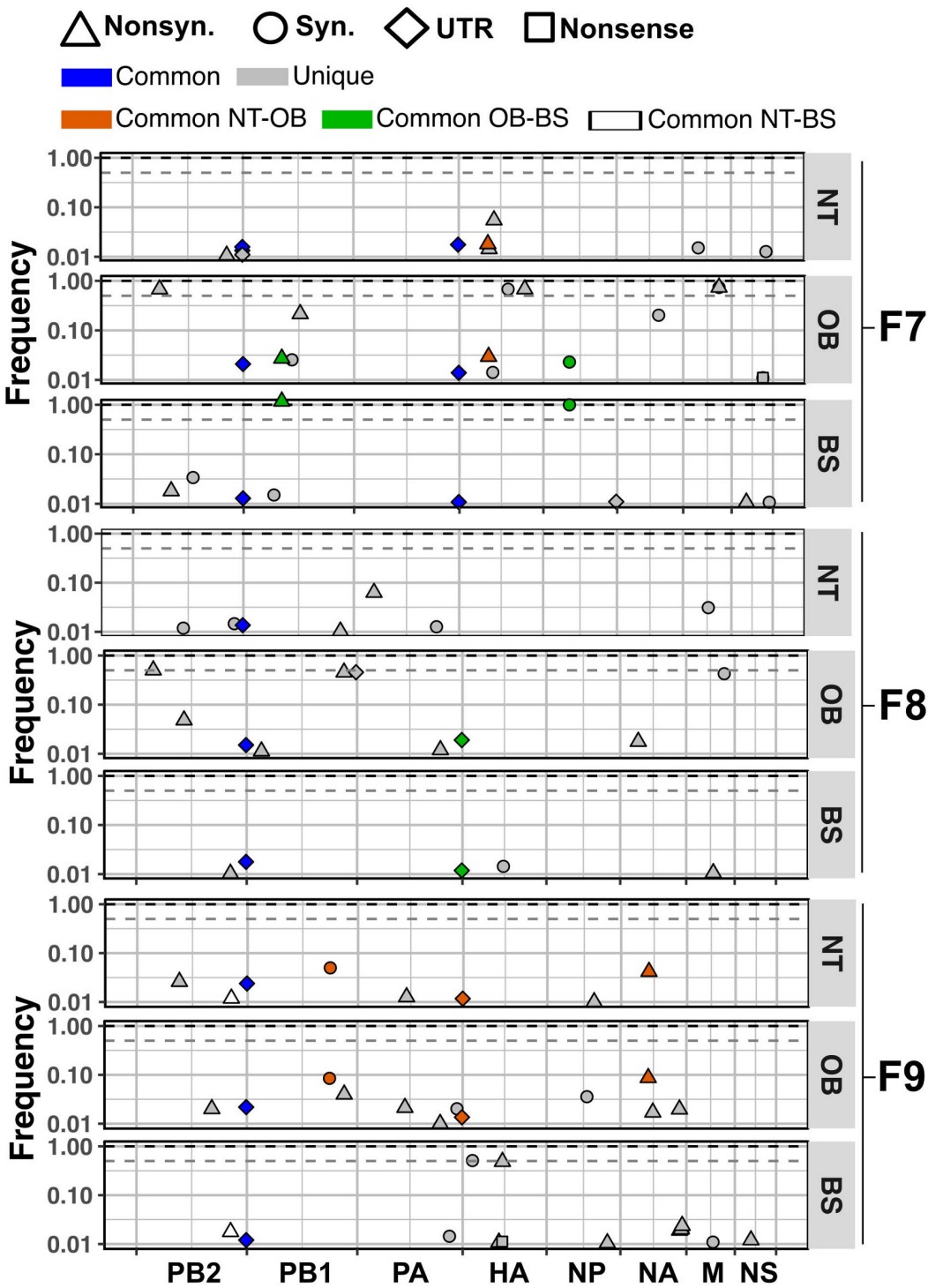

**Fig 6. Dispersion of H5N1-WT virus variants from nasal turbinates to the CNS.** Unique variants are presented in grey. Common variants between nasal turbinate (NT) and olfactory bulb (OB) are in orange. Common variants between the olfactory bulb and the brainstem (BS) are in green. Common variants between nasal turbinate and brainstem are in white. Variants that are common throughout are in blue. The types of mutation are represented by shapes: synonymous (Syn.) as circles, nonsynonymous (Nonsyn.) as tringles, variants in the untranslated regions (UTR) as diamonds and nonsense as squares. Nucleotide position within the concatenated viral genome is shown on the x-axis. Dashed grey line shows consensus cut off at frequency of 0.5 whereas the dashed black line marks a frequency of 1.

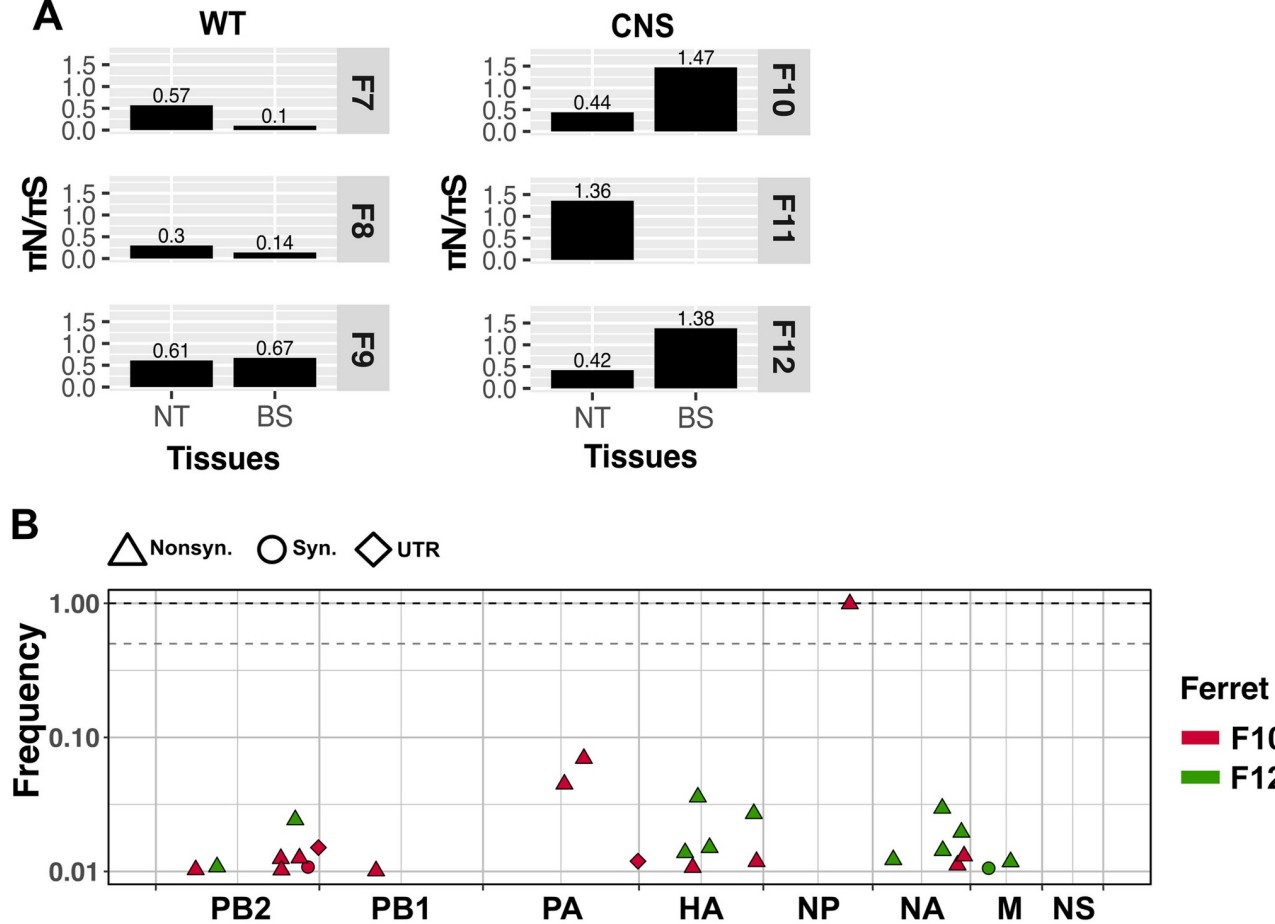

**Fig 7. Signs of positive selection for the H5N1-CNS virus are detected in the nasal turbinate and brainstem tissue of infected ferrets at 6 dpi.** Positive selection was calculated using the πN/πS ratio. Values above 1 denote positive selection (A). Variant analysis suggests independent evolutionary events shown by unique variants in brainstem tissue from ferrets F10 and F12 (B). The types of mutation are represented by shapes: synonymous (Syn.) as circles, nonsynonymous (Nonsyn.) as tringles and variants in the untranslated regions (UTR) as diamonds. The segments are shown as concatenated in the x-axis. Dashed grey line shows consensus cut off at 0.5 of frequency whereas dashed black line marks 1 of frequency.

on the H5N1-CNS virus in nasal turbinates in ferret F11 and brainstem tissues in ferrets F10 and F12 [Fig 7A]. To evaluate whether the signs of positive selection were the result of the selection of the same variants (parallel evolution) between individual ferrets, we compared the polymorphisms in virus populations present in the brainstem from ferrets F10 and F12 [Fig 7B]. We did not detect common variants between these samples, indicating that the positive selection detected in each animal led to independent paths of evolution.

## Discussion

In the present study we aimed to understand the basic mechanisms by which highly pathogenic avian influenza H5N1 viruses evolve once they infect the mammalian CNS. Using a ferret model, we found that influenza A/Indonesia/5/2005 (H5N1) virus was not subject to bottlenecks when entering the CNS via olfactory bulb, and that once inside the CNS, positive selection of variants can occur. Although the CNS-associated mutations examined did not appear to be adaptive, the viral dynamics observed suggest that an H5N1 virus can increase in fitness once it infects the CNS.

The substitutions PB1 E177G, PB1 A652T and NP I119M in the HPAI H5N1 virus background were found in a ferret with a severe meningo-encephalitis. Further evaluation of these substitutions revealed that they did not enhance virus spread to or viral replication in the CNS. In contrast, they appeared to restrict viral dissemination within infected ferrets, which might be related to a decreased fitness in the respiratory tract. The increased polymerase activity in cell culture and replication in MDCK cells could be associated with the location of PB1 E177G in the beta-ribbon domain, which has previously been implicated in increased polymerase activity in H5N1 viruses, possibly via interactions with host factors [27,28]. Similarly, the location of PB1 A652T suggests it could contribute to efficient stabilization of the 3' ssRNA positioning by interacting with a downstream base [22,23]. The apparent disconnection between the viral dissemination in vivo and the results from the cell culture-based assays likely relates to the difference in complexity of these systems. The increased in vitro polymerase activity implies that the CNS-associated mutations may have been fixed by positive selection but the fact that, in vivo, PB1 E177G reached fixation first while PB1 A652T seemed to offer greater advantage is possibly the result of the initial fixation of PB1 E177G and posterior selection of PB1 A652T.

Dispersion of virus populations to new anatomical sites can be restricted by physical barriers and/or limited capacity to replicate in the target tissue, which in turn can result in bottlenecks. For H5N1-WT virus, our analysis shows the absence of a bottleneck during virus spread from the nasal cavity to the CNS via the olfactory nerve, suggesting that this transfer occurs without restriction. Viruses can enter the olfactory bulb via the olfactory nerve through at least two mechanisms: i) infection of the olfactory receptor neurons, after which viruses are transported anterograde to the olfactory bulb; or ii) via diffusion through the channel formed by the olfactory ensheathing cells, which form an open connection to the CNS [2]. As H5N1 virus replicates efficiently in the olfactory mucosa it is likely that both mechanisms contribute. In contrast, influenza A viruses that do not replicate abundantly in the olfactory mucosa do not spread efficiently to the CNS via the olfactory nerve, like seasonal H3N2, 1918 pandemic H1N1 or 2009 pandemic H1N1 [7,29].

Even though in situ diversification seemed to be the dominant mechanism by which diversity was generated in viral populations, in H5N1-WT virus infected ferrets, dispersion of minor variants from the nasal turbinates to the CNS was detected at an appreciable frequency. The olfactory bulb received between 10 to 30% of the variants from the nasal turbinates whereas the brainstem received between ~7.5 to ~33%. The latter relationship suggests long range dissemination. Some variants were common across the three tissues, but some others were only shared between two sites. These data suggest that dispersion of some populations may follow the stepping stone model [30], in which populations migrate from the nasal turbinate to the olfactory bulb, diversify and then migrate to the brainstem. However, this mode of dispersion was not clear in all ferrets as the number and species of common variants varied between tissues. These differences can be explained by alternative points of access to the CNS such as the trigeminal ganglia.

In summary, here we show that an HPAI H5N1 virus can enter the CNS via the olfactory nerve without a genetic bottleneck, consistent with the high frequency of CNS involvement in mammals infected with these viruses. Efficient dispersal to the CNS via the olfactory nerve was associated with abundant infection and damage in the olfactory mucosa, which may explain reports of CNS infection by H5N1 viruses in the absence of overt respiratory clinical signs [31,32]. Importantly, we found that viral dynamics within the CNS are conducive to positive selection, giving the potential for improvement of viral replicative capacity, which may translate into more severe disease.

## Materials and methods

### Ethics statement

Research was conducted under a project license from the Dutch competent authority (license number AVD101002015340) and the study protocols were approved by the institutional Animal Welfare Body (Erasmus MC permit number 15-340-17).

### Viruses & virus titration

From a previous study in which ferrets were inoculated with HPAI A/Indonesia/5/05 (H5N1) virus (clade 2.1.3.2), a ferret with widespread CNS infection consistent with a severe meningo-encephalitis at 7 days post inoculation (dpi) was selected for further characterization [6]. From this ferret, viruses were sequenced from the nasal turbinates, cerebellum, and CSF and recombinant viruses were generated by reverse genetics. In brief, all eight influenza A virus gene segments were amplified by reverse transcription polymerase chain reaction (RT-PCR) and cloned in a modified version of the bidirectional reverse genetics plasmid pHW2000 as described before [33,34]. Substitutions of interest (PB1-E177G, PB1-A652T, NP-I119M) were introduced by site-directed mutagenesis using the QuikChange multi-site-directed mutagenesis kit (Stratagene, Leusden, Netherlands) according to the instructions of the manufacturer to generate the H5N1 viruses with single, double or triple mutants. The virus containing all three variants will be referred to as H5N1-CNS virus. Recombinant viruses were produced upon transfection of HEK293T cells as described previously [35]. Virus stocks were propagated once in Madin-Darby canine kidney (MDCK) cells and the presence of the substitutions of interest were verified using Sanger sequencing as described before [36]. The 50% tissue culture infectious dose (TCID50) in cell supernatant was determined by endpoint titration in MDCK cells, as described before [37]. All in vitro experiments involving HPAI H5N1 viruses were performed under biosafety level 3 conditions.

### Cells

Human neuroblastoma SH-SY5Y cells (Sigma-Aldrich,St. Louis, MO, USA) were maintained in a 1:1 mixture of Eagle minimal essential medium with Earle's Balanced Salt Solution (EMEM EBSS; Lonza, Breda, the Netherlands) and Ham's F-12 Nutrient Mixture (Thermo-Fisher Scientific, Waltham, MA, USA), supplemented with 10% fetal bovine serum (FBS; Sigma-Aldrich), 100 IU/ml penicillin (Lonza, Basel, Switzerland), 100 µg/ml streptomycin (Lonza), 2 mM glutamine (Lonza), 1.5 mg/ml sodium bicarbonate (Cambrex, Wiesbaden, Germany), sodium pyruvate (ThermoFisher Scientific) and 0.1 mM non-essential amino acids (MP Biomedicals Europe, Illkirch, France). Madin-Darby canine kidney cells (ATCC,Rockville, MD, USA) were maintained in EMEM supplemented with 10% FBS, 100 IU/ml penicillin, 100 µg/ml streptomycin, 2 mM glutamine, 1.5 mg/ml sodium bicarbonate, 1 mM, 10 mM HEPES (Cambrex), and 0.1 mM non-essential amino acids. Human lung epithelial carcinoma A549 cells (ATCC) were maintained in Ham's F-12 Nutrient Mixture (ThermoFisher), supplemented with 10% FBS, 100 IU/ml penicillin and 100 µg/ml streptomycin. HEK293T cells were cultured in Dulbecco's modified Eagle's medium (DMEM; Lonza) supplemented with 10% fetal calf serum (FCS), 100 IU/ml penicillin, 100 mg/ml streptomycin, 2 mM glutamine, 1 mM sodium pyruvate, and 0.1 mM non-essential amino acids.

### Polymerase activity assay

The polymerase acitivity (or minigenome) assay was performed as described previously [38], with the following alterations. The open reading frame of the polymerase basic protein 1

(PB1), polymerase basic protein 2 (PB2), polymerase acidic protein (PA) and nucleoprotein (NP) genes of A/Indonesia/5/2005 (H5N1) were cloned into the pPPI4 expression vectors [39] using Gibson assembly (New England Biolabs, Ipswich, MA, USA). Mutations in PB1 (E177G and A652T) and NP (I119M) were introduced using QuikChange II Site-Directed Mutagenesis Kit (Agilent, Santa Clara, CA, USA). HEK-293T, A549, and SH-SY5Y cells were seeded one day prior to the experiment into 96-well plates. Twenty-five ng of the firefly reporter plasmid, 50 ng of each of the plasmids encoding PB2, PB1, and PA, 100 ng of NP and 2 ng of the Renilla luciferase expression plasmid in 50 μl Opti-MEM (Gibco, Thermo Fisher) were mixed with 50 μl Opti-MEM containing Lipofectamine 2000 (Invitrogen, Thermo Fisher) for HEK-293T cells, A549 Cell Avalanche Transfection Reagent (EZ Biosystems, College Park, Maryland, United States) for A459 cells or SK-N-SH Cell Avalanche Transfection Reagent for SH-SY-5Y cells in a 1:3 ratio and incubated for 20 minutes at room temperature. Twenty μl of the transfection mixture was added to each well. Each transfection was performed in quadruplo in at least three independent experiments. Twelve- and 24-hours post transfection, luminescence was measured using the Dual-Luciferase Reporter Assay System (Promega) using a GloMax luminometer according to the manufacturer's instructions (Turner BioSystems). H5N1 PB2-627K served as an internal control for a "mammalian adapted" avian influenza virus polymerase complex with increased polymerase activity.

## Replication kinetics

Cells (MDCK, SH-SY5Y, and A549) were inoculated at a multiplicity of infection (MOI) of 0.001. After 1 hour of virus adsorption, cells were washed once and cultured in their respective serum-free medium in the absence of l-1-tosylamide-2-phenylethyl chloromethyl ketone (TPCK)-treated trypsin (Sigma-Aldrich). At 1, 6, 24, and 48 hours post infection (hpi), 100 μl supernatant was collected and stored at −80°C for subsequent virus end-point titration (see "Viruses & virus titration"). All experiments were performed three times independently with two technical replicates from which averages were used for statistical analysis.

## Structural modelling

The X-ray crystal structure of the influenza A virus polymerase heterotrimer A/duck/Fujian/01/2002(H5N1) in the RNA-free (apo) and conformation (PDB 6QPF) [23], nucleoprotein of A/WSN/1933 (PDB 2Q06)[40], and vRNP of A/Wilson-Smith/1933 (PDB 4BBL) [41] were used to map the locations of the mutations using the PyMOL Molecular Graphics System, Version 2.3 Schrödinger, LLC.

## Animal experiment

Ferrets were housed and experiments were performed in strict compliance with the Dutch legislation for the protection of animals used for scientific purposes (2014, implementing EU Directive 2010/63). Influenza virus and Aleutian Disease Virus seronegative 6-month-old female ferrets (*Mustela putorius furo*), were obtained from a commercial breeder (TripleF, USA). All animal experiments were performed under biosafety level 3+ conditions. The experiment in which the CNS-associated mutations were found was performed and described previously [6].

In total, twelve ferrets were divided into two groups of six ferrets each: one group inoculated with the H5N1-WT virus and the other group inoculated with the virus containing substitutions found in the CNS (PB1-E177G, PB1-A652T, NP-I119M; H5N1-CNS virus). A description of the ferret numbers and group assignment can be found in Table 3. On day 0, all ferrets were sedated with ketamine and medetomidine (antagonized with atipamezole) and inoculated intranasally with $10^6$ TCID50 of HPAI H5N1-WT or H5N1-CNS virus, divided over both nostrils (50 μL to

each nostril), and kept sedated for 10–15 minutes while on their backs, dorsal recumbency. Nasal and throat swabs were collected daily for virological analysis under ketamine sedation. Nasal swab samples were collected from the right nostril to keep the respiratory mucosa of the left nostril intact for pathological examination. Ferrets were weighed daily and observed for clinical signs according to the Reuman activity score [42]. At 3 and 6 days after inoculation, 3 randomly selected ferrets from each group were euthanized by exsanguination after anesthesia with ketamine and medetomidine, and tissues were collected for virological and/or pathological analysis, including nasal turbinates, trachea, lungs, tonsil, adrenal gland, tracheobronchial lymph node, liver, spleen, kidney, heart, pancreas, duodenum, jejunum, olfactory bulb, cerebrum, cerebellum, brainstem, cervical spinal cord, blood, and cerebrospinal fluid.

## Pathology and immunohistochemistry

All tissues collected during necropsy were fixed in 10% neutral-buffered formalin for ≥14 days. Tissues were embedded in paraffin, sectioned at 3 µm, and stained with hematoxylin-eosin for evaluation of histological lesions. For the detection of influenza virus antigen by immunohistochemistry, tissues were stained with a monoclonal antibody against influenza A virus nucleoprotein (clone HB-65; ATCC), as described elsewhere [43].

## RNA-seq library preparation and sequencing

RNA-Seq libraries were generated using the KAPA HyperPlus Kit (Roche, 0796248001) and KAPA Unique Dual-Indexed Adapter Kit (Roche, 08861919702), according to the manufacturer's protocol (Roche, SeqCap EZ HyperCap Workflow, version 2.2). In brief, all 8 gene segments of the influenza virus were amplified using Uni-12' and Uni-13' primers [44] specific for the 3' and 5' UTR with SuperScript III One-Step RT-PCR System with Platinum Taq High Fidelity DNA Polymerase (Invitrogen, 12574035). Cycling conditions were 55˚C for 2 min, 42˚C for 60 min, 94˚C for 2 min followed by 5 cycles of (94˚C for 30s; 44˚C for 30 s; 68˚C for 3.5 min) and 35 cycles of (94˚C for 30 s, 57˚C for 30 s; 68˚C for 3.5 min) and finally 68˚C for 10 min. Viral cDNA was purified using Agencourt AMPure XP beads (Beckman Coulter, A63880) and subjected to an enzymatic fragmentation step aimed at producing fragments of 300 bp. Following adaptor ligation, libraries were subjected to 7 cycles of PCR to produce libraries ready for sequencing with the following cycling conditions; 45 s 98˚C followed by 7 cycles (15 s 98˚C, 30 s at 60˚C, 30 s at 72˚C) and finally 1 min at 72˚C. Libraries were pooled in equimolar ratios, and sequencing (11 pM input concentration, spiked with 5% PhiX control V3 was performed using the Illumina Miseq (Miseq Control Software 2.6.2.1). The library pool was diluted and denatured according to the standard Miseq System Denature and Dilute Libraries Guide (Document # 15039740v10), and sequenced to generate paired-end 300 bp reads using a 600 cycle Miseq V3 reagent kit (illumina, MS-102-3003). After sequencing, demultiplexed fastq files were generated on the Miseq and analyzed using CLC Genomics Workbench (Qiagen, version 20) and pipelines developed in house.

## Variant analysis

Analysis of non-consensus variants was made using LoFreq [45] following the Genome Analysis Toolkit best practices [46]. After removing adapters using Cutadapt (version 2.8), reads were mapped back to their reference sequence using the option mem from BWA [47]. Data formatting for GATK was made using Picard (http://broadinstitute.github.io/picard/). The use of Mark-Duplicates from Picard was avoided as per LoFreq FAQs suggestion (https://csb5.github.io/lofreq/faq/) since samples were PCR products. Reads were realigned using RealignerTargetCreator and IndelRealigner from GATK. The quality of bases was recalculated using BaseRecalibrator

from GATK. The resulting bam file was used to perform variant calling analysis by LoFreq. Only variants at a frequency of 0.01 with a coverage equal or above 400 were used. For detection of synonymous and nonsynonymous mutations we used the program SNPdat [48].

### Diversity calculation

The π statistic for measuring nucleotide diversity was calculated using the synonymous (πS) and nonsynonymous (πN) nucleotide diversity using SNPGenie [49], which adapts Nei and Gojobori's (1986) method of estimating synonymous and nonsynonymous substitutions for next-generation sequencing data [50,51]. The cutoff used for calculation of π was set at a variant frequency of 0.01.

### Statistical analysis

Statistical analyses were performed using GraphPad Prism 8.2.1 software (La Jolla, CA, USA) for Mac. Each specific test is indicated in the figure legends. P-values of ≤0.05 were considered significant. All data are presented as means ± standard deviations (SD) or stand error of mean (SEM) indicated in the figure legend from at least three independent experiments.

### Plots

Some figures were made using the RStudio and the package ggplot2 [52] and aesthetically modified using Inkscape v0.48.1 (https://inkscape.org).

## Supporting information

**S1 Fig. Electrostatic surface potential of the polymerase and in proximity of PB1 residue 177 in apo-conformation.** Overview of the electrostatic surface potential of the polymerase complex (A) with closeup view of the PB1 at residue 177 with a glutamic acid (B) and glycine (C). Colors indicate a negative potential (red) and positive potential (blue) according to the color-coded electrostatic surface (unit KT/e).
(TIF)

**S2 Fig. Electrostatic surface potential of the polymerase and in proximity of PB1 residue 177 with bound vRNA promotor** (A). Overview of the electrostatic surface potential of the polymerase complex (B) with closeup view of the PB1 at residue 177 with a glutamic acid (C) and glycine (D). Colors indicate a negative potential (red) and positive potential (blue) according to the color scale bar.
(TIF)

**S3 Fig. Location of residue 652 in the priming loop of PB1.** (A), overview of the polymerase subdomains with PB1 (light-blue), PB2 (pale-yellow), and PA (pale-green), 5' RNA template, 3' RNA template, and mRNA product. (B), location of PB1 residue 651 (blue) and 652 (red) within the priming loop (light-blue). (C), location of PB1 residue 651 (blue) in contact (yellow dotted line) with the 3' RNA template.
(TIF)

**S4 Fig. Location of residue 119 in nucleoprotein.** (A), overview of the vRNP complex with NP (grey), RNA (black) and residue 119 (red). (B), the H5N1 NP (light-blue) superimposed on WSN NP (grey) with residue 119 (red) and RNA (black). (C), same as B, alternate view with closeup view displaying NP-interactions on opposite strands.
(TIF)

**S5 Fig. Electrostatic surface potential of the nucleoprotein and in proximity of residue 119.** Overview of the electrostatic surface potential of the polymerase complex (B) with closeup view (C & D) of the NP at residue 119 with an isoleucine (I) and methionine (M). Colors indicate a negative potential (red) and positive potential (blue) according to the color scale bar.
(TIF)

**S6 Fig. Virus detection in other organs of ferrets infected with H5N1-WT or H5N1-CNS virus, 3 and 6 dpi.** Dots represent individual ferrets while bars and lines represent means ±SDs respectively. Statistical analysis was performed using multiple independent unpaired t-tests. TCID; tissue culture infectious dose. Dotted lines represent the limit of detection.
(TIF)

**S7 Fig. Dispersion from nasal turbinates to the CNS in H5N1-CNS-infected ferrets.** Only samples from ferret that met cutoff are shown. Unique variants are presented in grey and common variants in blue. The types of mutation are represented by shapes: synonymous (Syn.) as circles, nonsynonymous (Nonsyn.) as tringles, variants in the untranslated regions (UTR) as diamonds and nonsense as squares. Nucleotide position within the concatenated viral genome is shown on the x-axis. Dashed grey line shows consensus cut off at frequency of 0.5 whereas the dashed black line marks a frequency of 1.
(TIFF)

**S8 Fig. Absence of positive selection in nasal turbinate at 3 dpi.** Bars and values above denote $\pi N/\pi S$ ratio.
(TIFF)

**S1 Table. In vivo stability of molecular signatures for CNS-mutations' sites at 3 dpi and alternative consensus alleles.**
(DOCX)

**S2 Table. In vivo stability of molecular signatures for CNS-mutations' sites at 6 dpi and alternative consensus alleles.**
(DOCX)

**S1 Data. Raw data mini genome assay.**
(XLSX)

**S2 Data. Raw data virus titration / growth kinetics.**
(XLSX)

**S3 Data. Raw data weight loss.**
(XLSX)

**S4 Data. Raw data viruses titration throat and nose swabs.**
(XLSX)

**S5 Data. Raw data virus titration organs.**
(XLSX)

## Acknowledgments

The authors would like to thank Prof. Ron Fouchier & Prof. Thijs Kuiken for the useful discussions.

## Author Contributions

**Conceptualization:** Jurre Y. Siegers, Lucas Ferreri, Thijs Kuiken, Anice C. Lowen, Sander Herfst, Debby van Riel.

**Data curation:** Jurre Y. Siegers, Lucas Ferreri.

**Formal analysis:** Jurre Y. Siegers, Lucas Ferreri, Dirk Eggink, Edwin J. B. Veldhuis Kroeze, Aartjan J. W. te Velthuis, Debby van Riel.

**Funding acquisition:** Sander Herfst, Debby van Riel.

**Investigation:** Jurre Y. Siegers, Lucas Ferreri, Dirk Eggink, Edwin J. B. Veldhuis Kroeze, Aartjan J. W. te Velthuis, Marco van de Bildt, Lonneke Leijten, Peter van Run, Dennis de Meulder, Theo Bestebroer, Mathilde Richard, Anice C. Lowen, Sander Herfst, Debby van Riel.

**Methodology:** Jurre Y. Siegers, Lucas Ferreri, Dirk Eggink, Edwin J. B. Veldhuis Kroeze, Aartjan J. W. te Velthuis, Lonneke Leijten, Peter van Run, Theo Bestebroer, Mathilde Richard, Thijs Kuiken, Anice C. Lowen, Sander Herfst, Debby van Riel.

**Project administration:** Debby van Riel.

**Resources:** Aartjan J. W. te Velthuis, Debby van Riel.

**Supervision:** Aartjan J. W. te Velthuis, Anice C. Lowen, Debby van Riel.

**Visualization:** Jurre Y. Siegers, Lucas Ferreri, Debby van Riel.

**Writing – original draft:** Jurre Y. Siegers, Lucas Ferreri, Anice C. Lowen, Debby van Riel.

**Writing – review & editing:** Jurre Y. Siegers, Lucas Ferreri, Mathilde Richard, Anice C. Lowen, Sander Herfst, Debby van Riel.

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
