## [Decision Letter · Decision Letter 0]

27 Oct 2022

Dear Dr van Riel,

Thank you very much for submitting your manuscript "Evolution of highly pathogenic H5N1 influenza A virus in the central nervous system of ferrets" for consideration at PLOS Pathogens. As with all papers reviewed by the journal, your manuscript was reviewed by members of the editorial board and by several independent reviewers. In light of the reviews (below this email), we would like to invite the resubmission of a significantly-revised version that takes into account the reviewers' comments.

The concerns raised by Reviewer 2, in particular, should be carefully considered including how to address the concern that the CNS-associated virus mutations do not cause the same phenotype in ferrets. The biological relevance of the virus mutations must be addressed.

We cannot make any decision about publication until we have seen the revised manuscript and your response to the reviewers' comments. Your revised manuscript is also likely to be sent to reviewers for further evaluation.

Sincerely,

Sabra L. Klein

Associate Editor

PLOS Pathogens

Ana Fernandez-Sesma

Section Editor

PLOS Pathogens

Kasturi Haldar

Editor-in-Chief

PLOS Pathogens

orcid.org/0000-0001-5065-158X

Michael Malim

Editor-in-Chief

PLOS Pathogens

orcid.org/0000-0002-7699-2064

The concerns raised by Reviewer 2, in particular, should be carefully considered including how to address the concern that the CNS-associated virus mutations do not cause the same phenotype in ferrets. The biological relevance of the virus mutations must be addressed.

Reviewer's Responses to Questions

**Part I - Summary**

Reviewer #1: Siegers et al have performed detailed experiments to explore the mechanisms by which the HPAI H5N1 virus evolves in the mammalian CNS. The authors demonstrate that an HPAI H5N1 virus can enter the CNS via the olfactory nerve without a genetic bottleneck. And the authors conclude that viral dynamics within the CNS are conducive to positive selection, which gives the potential for improvement of viral replicative capacity and may lead to more severe disease.

Reviewer #2: Professor van Riel and colleagues report findings from in vitro and in vivo studies that examined the contribution of specific amino acid residues of an H5N1 influenza A virus strain to CNS invasion and neurological disease. Cell culture studies revealed that the CNS-associated mutations enhanced the in vitro viral replication phenotype. Follow-up analysis with the ferret model of influenza and an H5N1 influenza virus strain harboring these putative CNS-associated mutations revealed an attenuated viral phenotype, including restricted tissue distribution. The results suggest the consequence of positive selection on viral phenotypes and potential evolution of influenza viruses.

**Part II – Major Issues: Key Experiments Required for Acceptance**

Reviewer #1: 1.Lines 104-111. Please explain why the change in the surface potential from negative to neutral in the apo form and from moderately negative to moderately positive charge in the promotor-bound form influence interactions with potential (host) binging factors?

2. Lines 133-137. Both the single substitution PB1-652T and the combined substitutions PB1-652T/177G with NP-119M can result in a significant increase of polymerase activity in A549 cells. However, the combination of PB1-652T with NP-119M maintained wild-type levels of polymerase activity. Please give some reasonable explanation for these phenotypes that seem contradictory.

Reviewer #2: Genetic analysis of the viral population within the nasal cavity did not reveal a potential bottleneck or selection of virus with CNS-associated mutations. These results suggest the possibility that viral quasispecies contribute to the observed CNS invasive phenotype, neurological disease, and broad tissue distribution, which was lost by selection of a specific viral strain with the CNS-associated mutations. Another limitation of the study is that this phenotype was observed for one infected ferret. It is unclear how the host-specific response of this one animal contributed to the observed phenotype and if the selective forces are reproducible in shaping evolution of influenza viruses.

**Part III – Minor Issues: Editorial and Data Presentation Modifications**

Reviewer #1: 1.It is better to specify which amino acids were site-directed mutated to generate the H5N1-CNS virus in Materials and methods or anywhere proper.

2.Line 125. Where are the large positively charged RNA-biding grove? Please highlight it in the supplementary figure 5.

3. Line 127. “different ‘human’ cell types” is suggested for more specific description.

4.Fig 5. Whether the difference of genetic diversity is significant statistically?

Line 205.

5.Line 205. The study aimed to explore the CNS-associated mutations of an H5N1 HPAIV, did not involve reassortment events among different viruses, so “variant” may be appropriate than “genotypes”.

6.Lines 222, 223. Please provide more information to explain what the sentence “it is highly unlikely for common variants penetrated the CNS from the nasal turbinate” .

7.Line 278. what’s means of “stochastic fixation”？

8. Statistical method in Fig. 2. Whether one-way ANOVA should be used instead of two-way ANOVA?

Reviewer #2: not-applicable

PLOS authors have the option to publish the peer review history of their article (what does this mean?). If published, this will include your full peer review and any attached files.

Reviewer #1: **Yes: **Honglei Sun

Reviewer #2: No
---

## [Decision Letter · Decision Letter 1]

16 Feb 2023

Dear Dr van Riel,

We are pleased to inform you that your manuscript 'Evolution of highly pathogenic H5N1 influenza A virus in the central nervous system of ferrets' has been provisionally accepted for publication in PLOS Pathogens.

Best regards,

Sabra L. Klein

Academic Editor

PLOS Pathogens

Ana Fernandez-Sesma

Section Editor

PLOS Pathogens

Kasturi Haldar

Editor-in-Chief

PLOS Pathogens

orcid.org/0000-0001-5065-158X

Michael Malim

Editor-in-Chief

PLOS Pathogens

orcid.org/0000-0002-7699-2064

Reviewer Comments (if any, and for reference):

Reviewer's Responses to Questions

**Part I - Summary**

Reviewer #1: The manuscript modification meets the requirement.

Reviewer #2: This in vivo study examined the replication phenotype and tissue distribution of H5N1 highly pathogenic avian influenza virus in the ferret model of influenza. The authors note that the findings are limited to one animal but the data indicate the impact of genetic evolution for virus replication and viral pathogenesis. Overall, the authors deserve high marks for execution and scholarship.

**Part II – Major Issues: Key Experiments Required for Acceptance**

Reviewer #1: No

Reviewer #2: Not applicable.

**Part III – Minor Issues: Editorial and Data Presentation Modifications**

Reviewer #1: No

Reviewer #2: Not applicable.

PLOS authors have the option to publish the peer review history of their article (what does this mean?). If published, this will include your full peer review and any attached files.

Reviewer #1: No

Reviewer #2: No

---

## [Editor Report · Acceptance letter]

3 Mar 2023

Dear Dr van Riel,

We are delighted to inform you that your manuscript, "Evolution of highly pathogenic H5N1 influenza A virus in the central nervous system of ferrets," has been formally accepted for publication in PLOS Pathogens.

Best regards,

Kasturi Haldar

Editor-in-Chief

PLOS Pathogens

orcid.org/0000-0001-5065-158X

Michael Malim

Editor-in-Chief

PLOS Pathogens

orcid.org/0000-0002-7699-2064